# Enhanced Thermal Conductivity of Silicone Composites Filled with Few-Layered Hexagonal Boron Nitride

**DOI:** 10.3390/polym12092072

**Published:** 2020-09-12

**Authors:** Wei-Cheng Cheng, Yi-Ting Hsieh, Wei-Ren Liu

**Affiliations:** Department of Chemical Engineering, R&D Center for Membrane Technology, Research Center for Circular Economy, Chung Yuan Christian University, 200 Chung Pei Road, Chung Li District, Taoyuan City 32023, Taiwan; z902082005@gmail.com (W.-C.C.); sunday52052@gmail.com (Y.-T.H.)

**Keywords:** hexagonal boron nitride, jet cavitation, silicone, thermal conductivity, silicone

## Abstract

In this study, we demonstrate the use of silicone/few-layered hexagonal boron nitride (FL-hBN) composites for heat dissipation applications. FL-hBN is synthesized via a green, facile, low-cost and scalable liquid exfoliation method using a jet cavitation process. The crystal structures, surface morphologies and specific surface areas of pristine h-BN and FL-hBN were characterized by XRD, SEM, TEM and AFM (atomic force microscopy). The results confirmed that FL-hBN with a thickness of ~4 nm was successfully obtained from the exfoliation process. In addition, we introduced both pristine h-BN and FL-hBN into silicone with different ratios to study their thermal properties. The results of the laser flash analysis indicate that the silicon/FL-hBN composite exhibited a higher thermal conductivity than that of the silicone/h-BN composite. With the optimal loading content of 30 wt.% FL-hBN content, the thermal conductivity of the composite could be enhanced to 230%, which is higher than that of silicone/h-BN (189%). These results indicate that jet cavitation is an effective and swift way to obtain few-layered hexagonal boron nitride that could effectively enhance the thermal conductivity of silicone composites.

## 1. Introduction

In recent decades, heat dissipation has been one of the most critical challenges in current high-density and high-power electronic products due to rapid developments in the miniaturization of microelectronic devices. Silicone-based materials have been widely applied as the preferred matrix material, such as elastomeric thermal pads [1] and electronic packaging materials [2], due to their good electrical insulation, thermal stability, superior adhesion, good mechanical properties, ease of processing and low cost. However, it is well known that silicone has some disadvantages, such as a complex packaging process and low thermal conductivity, which might limit its application in the advanced microelectronic packaging field [3,4]. Hence, the development of highly thermally conductive silicone-based composites which maintain their low dielectric constant plays an important role in electronic packaging technologies.

It is well known that some potential fillers with high thermal conductivity material, such as graphene [5,6], metal particles [7,8], aluminum nitride (AlN) [9,10], alumina (Al_2_O_3_) [11,12] and silicon carbide (SiC) [13,14], have been reported. Mixing these fillers into polymer matrix composites could greatly increase the thermal conductivity of the polymer composites. In recent years, there have been many reports about improving the thermal conductivity of polymer matrices by incorporating hexagonal boron nitride (h-BN) due to its excellent properties, including high thermal conductivity, electrical insulation and excellent mechanical properties [15,16]. Hou’s group grafted silane molecules onto the surface of BN particles to improve the wettability and homogeneous dispersion of BN in the polymer matrix with a strong interface interaction. A thermal conductivity of 1.178 W/m·K was obtained at 30 wt.% modified-BN loading [17]. Muratov et al. focused on the investigation of hexagonal boron nitride powder (h-BN) in combination with 3-amino-propyl-3-ethoxy-silane (APTES) as a thermal conductivity-increasing filler for polypropylene. They discovered that annealed h-BN powder before treatment by APTES had the highest –OH group. The maximum filler load increased from 11.6 to 33.7 wt.%, and the thermal conductivity of the composite samples produced using surface treated filler powder increased from 0.256 to 0.369 W/m·K at room temperature [18]. According to these references, we could understand that by introducing modified-BN as a filler into a polymer matrix, the thermal conductivity could be enhanced. When the filler in a polymer matrix has a larger aspect ratio, the thermal conductivity of the composite will be more enhanced based on the same weight percentage because fillers with a large aspect ratio easily form bridges between each other [19,20].

The aspect ratio is given by the ratio of the diameter of the platelet to its laminar thickness [21]. Thus, we can obtain a large aspect ratio if we decrease the thickness of filler. According to the existing research, the preparation of two-dimensional nanomaterials generally includes a micro-mechanical exfoliation method [22], ball milling [23,24], laser exfoliation [25], liquid exfoliation [26,27,28], a lithium intercalation method [29] and chemical vapor deposition (CVD) [30,31]. Fan et al. demonstrated a novel approach to fabricate boron nitride nanosheets (BNNS) via hypochlorite-assisted ball milling. This method involves the synergetic effects of chemical peeling and mechanical shear forces, which can improve the yield and dispersion [32]. However, the disadvantage of this process is that it would generate many defects due to the strong hit in the process of ball milling. Lithium intercalation is another approach to prepare few-layered materials. The drawbacks of this approach, however, include its complicated process and incomplete removal of lithium ions in the intercalation process. Thus, this process cannot be practically applied. Recently, liquid exfoliation methods have been reported as simple and solution-processable methods for preparing few-layered nanosheets. In this process, ultrasonic waves are applied to a mixture of bulk layered material and an appropriate solvent is chosen; the exfoliated few-layered nanosheets are produced by enough energy to overcome the Van der Waals forces between layers. Wu et al. obtained a thickness of BNNS (boron nitride nanosheets) between one to three layers using the liquid exfoliation method. Experimental results indicated that the BNNS could improve thermal stability and promote the curing of the matrix [33]. Recent studies have used liquid exfoliation techniques of two-dimensional materials, such as high pressure homogenization [34] and sonication [35], to delaminate these materials, or used a jet cavitation method to obtain few-layer two-dimensional materials [36]. According to the above research, these solvents are organic-based solvents, such as *N*-Methyl Pyrrolidone (NMP), Dimethylformamide (DMF) or acetone, which are used in the preparation process, and even the surfactants, including sodium carboxymethyl cellulose (CMC), non-ionic surfactant TWEEN^®^80 (TW80) or sodium dodecyl sulfate (SDS), are used. In our previous work, we synthesized few-layer graphene, Tungsten disulfide (WS_2_) and Molybdenum diselenide (MoSe_2_) by using a jet cavitation method [37,38,39].

In this study, few-layered hexagonal boron nitrides (FL-hBN) were firstly obtained via a jet cavitation method. The as-synthesized and the pristine FL-hBN were used as fillers in silicone to enhance its thermal conductivity. The jet cavitation method is a rapid, environmental-friendly, cost-effective and facile process. In order to prove that the FL-hBN was successfully obtained, h-BN and FL-hBN were characterized by using SEM, AFM (atomic force microscopy) and TEM to observe the thickness and testing BET (Brunauer–Emmett–Teller) to obtain the specific surface areas to calculate aspect ratios. In the thermal property analysis, the silicone composites with FL-hBN showed a much higher thermal conductivity than that of the h-BN/silicone composite in the same loading weight.

## 2. Experimental Section

### 2.1. Preparation of FL-hBN Powder

1 g hexagonal boron nitride (h-BN, purity > 99%, Asia Carbons & Technology lnc.^®^, Taoyuan, Taiwan) was mixed with 100 mL of deionized water to achieve a concentration of 1 wt.%. A dispersion of h-BN in H_2_O was transferred into the tank of the low temperature ultra-high pressure continuous homemade flow cell disrupter (LTHPD). The solution was poured into the device and the process was operated three times at different pressures (800, 1300 and 1800 bar) in a circulation cooling water bath which kept the temperature at 14–16 °C. Afterwards, the few-layered h-BN suspension was produced by high pressure (Appendix A). Finally, the black particles were collected by centrifugation (10000× *g* rpm) and washed with ethanol several times. After drying at 80 °C, FL-hBN powder was obtained.

### 2.2. Preparation of h-BN/silicone and FL-hBN/silicone Composites

A schematic diagram of the preparation of the h-BN/silicone and FL-hBN/silicone composites is shown in Figure 1. Firstly, 3 g silicone and different weight ratios of h-BN (10, 20 and 30 wt.%) and FL-hBN (10, 20 and 30 wt.%) were mixed well in the beaker. Appropriate amounts of ethyl ethanoate (Sigma Aldrich^®^, St. Louis, MO, USA, purity > 99.8%) were introduced into the beaker with magnetic stirring for 5 h. Secondly, the mixing solution was put in the furnace at 150 °C for 1 h to evaporate the solvent. Third, 3 g hardener (Sigma Aldrich^®^, KJC-1200) was added into the beaker and mixed for 2 h. Finally, we put the final product into the mold at 150 °C in a vacuum for thermal curing. The as-prepared bare silicone, h-BN/silicone and FL-hBN/silicone composites were obtained.

## 3. Characterizations

The thermal behaviors of the as-prepared materials were investigated using TGA analysis (thermogravimetric analysis). The ramp rate was maintained at 5 °C/min under air atmosphere. The crystal phase structure and purity were determined by an XRD (X-ray diffractometer) analysis using a D8 diffractometer (Bruker^®^, Billerica, MA, USA) with monochromatic CuKα radiation. The operating voltage, current and wavelength (λ) were 40 kV, 30 mA and 1.54060 Å, respectively. Diffraction data were recorded in the range (2θ) of 10°–80°. The morphological natures of the as-prepared samples were observed using SEM (scanning electron microscopy, Hitachi S-4100, Okinawa, Japan) with electron mapping (EDS, energy-dispersive X-ray spectroscopy, Okinawa, Japan). Atomic force microscope (AFM) images were captured by a Bruker Dimension Icon. The samples for AFM were prepared by dropping the dispersion directly onto freshly cleaved mica wafers with an injector. The thermal diffusivity (α) of the film was measured by a laser flash thermal analyzer (LFA457 Micro Flash, Netzsch, Germany). The sample size was 1 cm × 1 cm × 0.2 cm. Brunauer–Emmett–Teller (BET) specific surface area was determined from N_2_ adsorption by using a Micromeritics TriStar 3000 (Norcross, GA, USA) analyzer at liquid nitrogen temperature.

## 4. Results and Discussion

The XRD patterns of pristine h-BN and few-layered h-BN (FL-hBN) are shown in Figure 2a,b. All the diffraction peaks at 2θ values of 26.76°, 41.59°, 43.87°, 50.15° and 55.17° corresponded to the diffraction planes of h-BN in (002), (100), (101), (102) and (004), respectively. All of the diffraction peaks matched well with the standard values and agreed with the hexagonal structure of the Bragg positions in JCPDS-34-0421, shown in Figure 2c. Compared with the pristine h-BN, the (002) and (004) peaks of few-layer BN showed a remarkable peak broadening, indicating the presence of thinner h-BN sheets and much less extended/ordered stacking in the c direction (Appendix A). Figure 2d shows photo images of the h-BN and FL-hBN solutions with different settling times. Obviously, after the exfoliation process, the settling time of FL-hBN was much longer than that of pristine h-BN, which might be due to the fact that the thickness of h-BN may have become much thinner after the exfoliation process.

Figure 3a,d show the FE-SEM images of the h-BN and FL-hBN, respectively. The lateral size of h-BN was 4–9 μm. After the exfoliation process, the lateral size of the FL-hBN was smaller than that of the h-BN in the region of 2–5 μm. Figure 3a,b also demonstrated that the thickness of the FL-hBN prepared by a jet cavitation method was thinner than that of h-BN. The phenomenon proved that h-BN could be exfoliated successfully. Figure 3b,e display TEM images of the h-BN and FL-hBN. The FL-hBN became transparent due to the small thickness shown in Figure 3e. The HR-TEM images of these samples are also shown in Figure 3c,f. A lattice fringe with interplane spacing of 1.67 Å, corresponding to (004) plane, was revealed. These results could be seen due to the change in thickness with FL-hBN after delamination.

The surface morphology and thickness distribution of the as-synthesized FL-hBN was further examined by AFM characterization, as shown in Figure 4a, by 30 samples. The results indicate that the thickness of the h-BN was ~500 nm (Figure 4a), which is much thicker than that of the FL-hBN with a thickness in the range of 3.9 nm~4.3 nm, shown in Figure 4b. Figure 4c shows the histogram of the thickness distribution from AFM images of the FL-hBN from 30 samples. The results further confirm that more than 40% of the FL-hBN nanosheets’ thicknesses ranged between 3.8 and 4.2 nm.

When the bulk material is peeled off layer by layer, it can be imagined that the specific surface area will change. Thus, we used N_2_ adsorption and desorption isothermal curves of h-BN (black line + symbol) and FL-hBN (red line + symbol), shown in Figure 5, to understand the difference in specific surface areas. These two samples demonstrated type IV isotherms, as determined by the International Union of Pure and Applied Chemistry (IUPAC), which is usually associated with mesoporous materials (i.e., pore sizes between 2 and 50 nm). The specific surface areas of the h-BN and FL-hBN were about 4.456 and 11.19 m^2^g^−1^, respectively. In addition, from the data of pore size distribution analysis, the corresponding pore volumes of h-BN and FL-hBN were 0.01917 and 0.04241 cm^3^g^−1^, respectively. The higher specific surface area of FL-hBN could represent that the method we used to exfoliate materials was successful.

Figure 6 shows the XRD analysis of the composites after the h-BN or FL-hBN were added into the silicone. The lattice orientation of the filler in the composites can be observed. It can be proved that h-BN exhibited high crystallinity in the composites (h-BN/silicone). Highly crystalline materials may have some advantages in some properties, such as their mechanical properties that are mentioned later. The diffraction intensity of FL-hBN on (002) and (004) decreased in composites with FL-hBN, indicating that the thickness of the filler (FL-hBN) was thinner in the composite. This trend was consistent with other reports [40,41,42,43].

Figure 7a,b show the thermal diffusivity and thermal conductivity of the composites of h-BN/silicone and FL-hBN/silicone composites with different loading ratios. Table 1 summarizes the physical properties of all of these composites. The results showed that both thermal diffusivity and thermal conductivity increased with the increasing content of the fillers. The composites containing FL-hBN exhibited much higher thermal conductivity than those with pristine h-BN at the same fraction in silicone. The thermal conductivity of 30% FL-hBN/silicone (30 wt.% FL-hBN into silicone) composites was 0.515 W/m·K, which was higher than that of 30% hBN/silicone composites (0.424 W/m·K). Compared to pristine silicone, the thermal conductivity of the FL-hBN/silicone composite enhanced by 230% (2.3 times) by introducing 30 wt.% FL-h-BN.

The aspect ratio (Equation (1)) of the filler affects the thermal properties of composites [21]. Materials possessing a higher aspect ratio could disperse well in a polymer matrix and build a good thermal conductive network in the composite even if the filler content is low. We can obtain the aspect ratio from the measured specific surface area by BET and the particle size of the filler. The aspect ratios of h-BN and FL-hBN were 29.06 and 56.10, respectively. From these results, FL-hBN has a larger aspect ratio so it can enhance compatibility with the polymer matrix. In order to reveal the filler dispersing in these composites, the morphologies of the composites were characterized by SEM micrographs in Appendix A. We can observe that FL-hBN had excellent dispersion in the composites of different filler loading. Compared with h-BN/silicone, h-BN could not connect well in the polymer matrix. The same results were shown in EDX mapping (Figure 8a,b).
(1)$$\mathrm{AR}\;=\;\frac{D}{t}=\frac{S\rho D}{2}-2$$
whereAR = Aspect ratio,D = The diameter of the platelets (average particle size),t = The thickness of the platelets,S = The specific surface area of the particles,ρ = The density of the platelets.

Figure 9 shows the TGA curves for pure silicone, h-BN/silicone (30 wt.%) and FL-hBN/silicone (30 wt.%) at 800 °C in nitrogen. The results showed that the temperature for 5% weight loss of these composites appeared at 410.1 °C (bare silicone), 483.1 °C (h-BN/silicone (30 wt.%)) and 488.4 °C (FL-hBN/silicone (30 wt.%)). At 600 °C, bare silicone, h-BN/silicone (30 wt.%) and FL-hBN/silicone (30 wt.%) exhibited 19.67%, 16.54% and 13.06% weight loss, respectively, indicating that the FL-hBN/silicone composite had lower weight loss than the other. If the fillers connect well with each other in the polymer matrix, heat will transfer quickly by thermal conduction. On the other hand, FL-hBN in silicone prevented heat from accumulating in the polymer matrix so that the composite will decompose more slowly.

Dynamic mechanical analysis (DMA) was performed to obtain the temperature dependent properties of composites, such as the storage modulus. The storage modulus is highly dependent on the dispersion between the filler and the silicone. Figure 10 shows the DMA curve of the three composites. The storage modulus of the composites increased after adding the fillers. For example, at 328 °C, the storage modulus of the pure silicone and the composites with hBN and FL-hBN were 9.19, 20.73 and 19.86 MPa, respectively. The main reason for the higher mechanical property is that the fillers will transfer some strength to each other when composite is stretched by external force.

## 5. Conclusions

We successfully obtained FL-hBN with an extremely facile and environmental-friendly jet cavitation method. According to AFM analysis, the thickness of all the FL-hBN layers were less than 5 nm. Moreover, the FL-hBN layers had smaller particle sizes with an increased surface area and a higher aspect ratio of 56.10. The thermal properties of both h-BN/silicone and FL-hBN/silicone composites improved, indicating that hexagonal boron nitride is an exceptional filler for silicone. Silicone composites containing FL-hBN exhibited both outstanding thermal conductivity and thermal stability—better than composites with h-BN. Compared to pure silicone, the thermal conductivity of the FL-hBN/silicone composite was 0.515 W/m·K at 30 wt.% filler loading. Silicone incorporated with FL-hBN also exhibited higher thermal stability at 800 °C. FL-hBN has a higher aspect ratio, indicating that it has good dispersion in silicone rubber, so the filler can connect well to form a thermal network. However, exfoliating h-BN to FL-hBN yielded superior thermal conductivity properties which can be used for industrial applications.

## Figures and Tables

**Figure 1 polymers-12-02072-f001:**
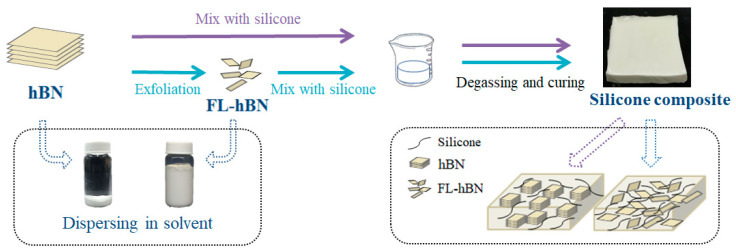
Schematic diagram of the preparation of hexagonal boron nitride (h-BN)/silicone and few-layered hexagonal boron nitride (FL-hBN)/silicone composites by our approach.

**Figure 2 polymers-12-02072-f002:**
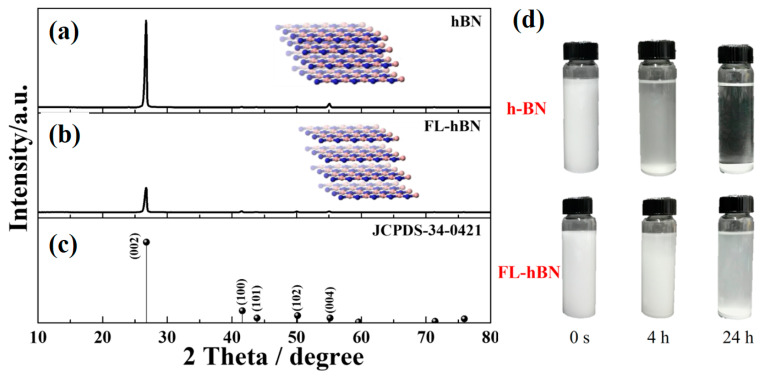
XRD patterns of: (**a**) h-BN; (**b**) FL-hBN; (**c**) standard data JCPDS-34-0421. Photo images (**d**) of h-BN and FL-hBN solutions with different settling times.

**Figure 3 polymers-12-02072-f003:**
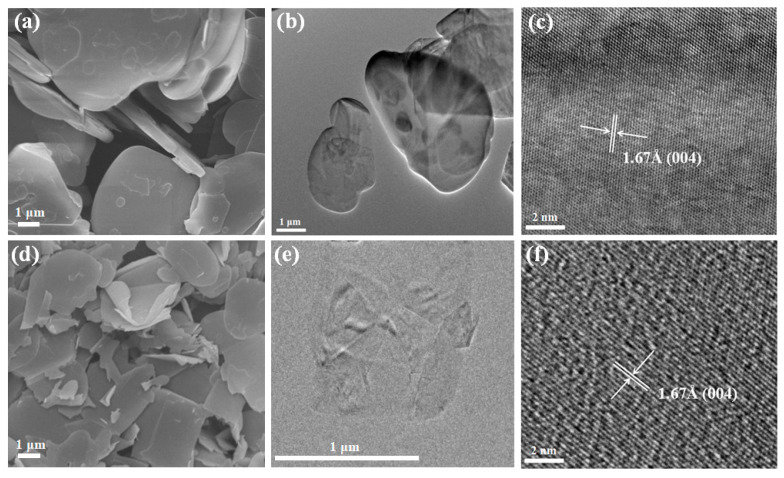
SEM images of (**a**) h-BN and (**d**) FL-hBN; TEM images of (**b**) h-BN and (**e**) FL-hBN; HR-TEM images of (**c**) h-BN and (**f**) FL-hBN. The insets show the selected area electron diffraction patterns.

**Figure 4 polymers-12-02072-f004:**
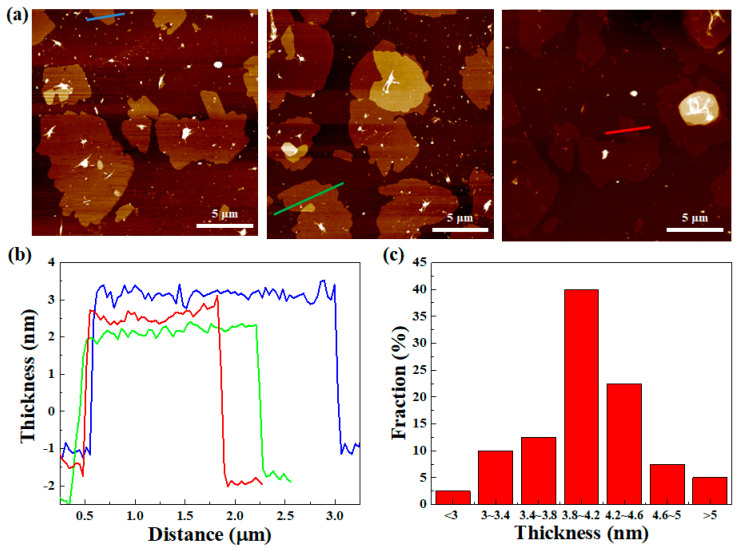
(**a**) AFM images (**b**) and the corresponding line scans of the FL-hBN on Si substrate; (**c**) histogram of the number visual observations of sheets as a function of the number of the layers of FL-hBN per sheet.

**Figure 5 polymers-12-02072-f005:**
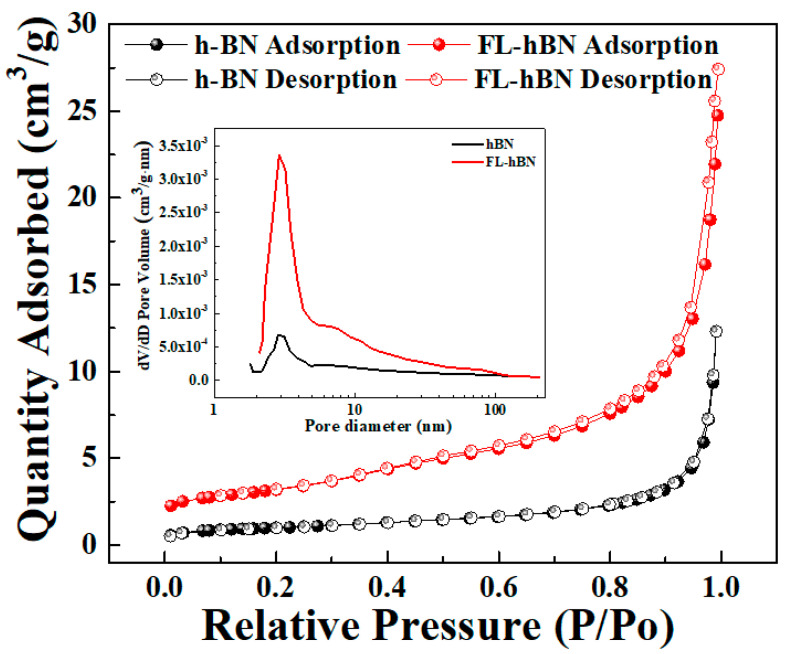
N_2_ adsorption/desorption isotherms and pore size distribution analyses of h-BN (black line + symbol) and FL-hBN (red line + symbol). Inset: pore size distribution analyses of h-BN and FL-hBN.

**Figure 6 polymers-12-02072-f006:**
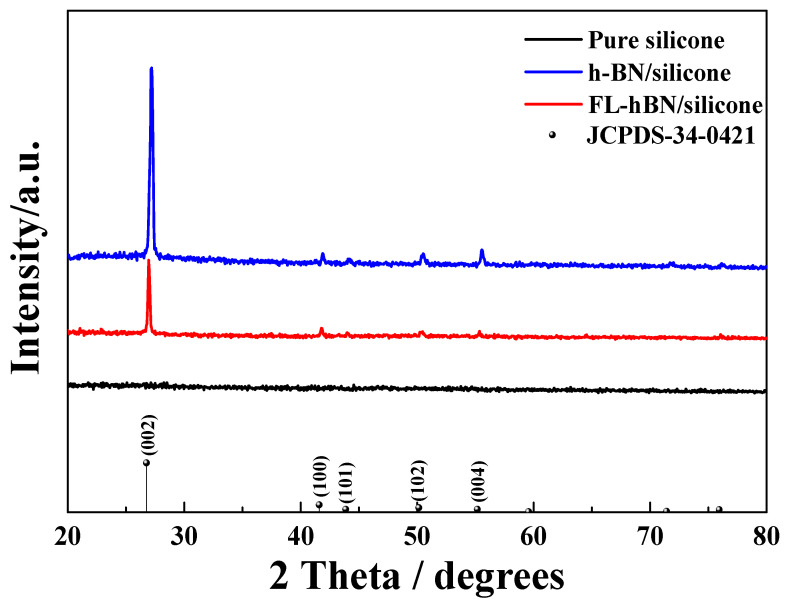
XRD patterns of pure silicone, (30%) h-BN/silicone and (30%) FL-hBN/silicone.

**Figure 7 polymers-12-02072-f007:**
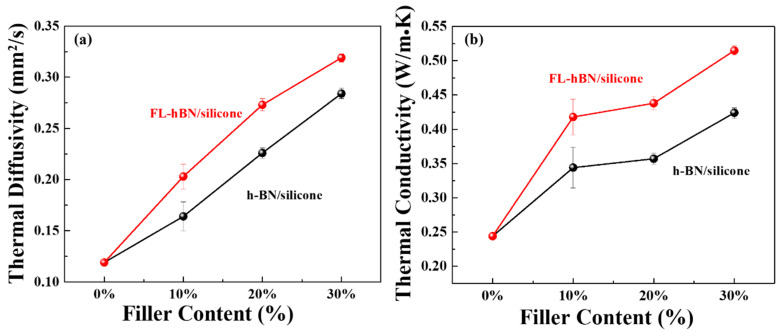
(**a**) Thermal diffusivity and (**b**) thermal conductivity of h-BN/silicone composites and FL-hBN/silicone composites.

**Figure 8 polymers-12-02072-f008:**
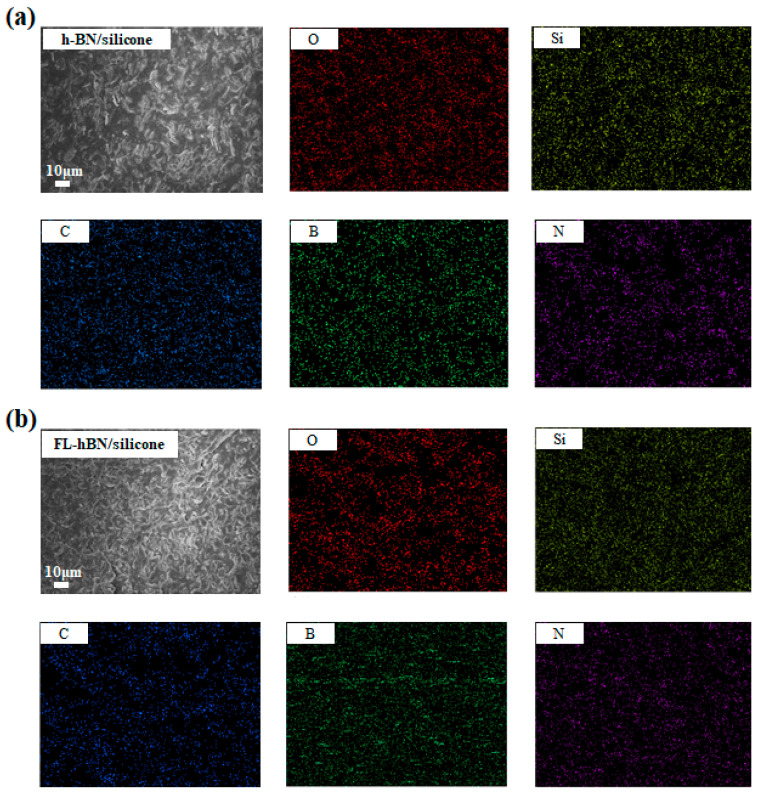
The Energy-dispersive X-ray spectroscopy (EDX) mapping of (**a**) h-BN/silicone (30 wt.%) and (**b**) FL-hBN/silicone (30 wt.%).

**Figure 9 polymers-12-02072-f009:**
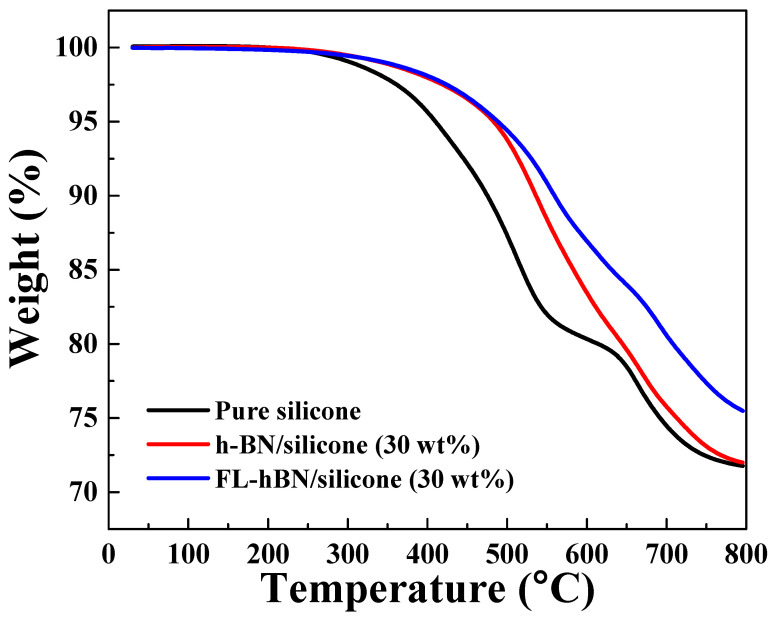
Thermal stabilities of pure silicone, hBN/silicone composites and FL-hBN/silicone.

**Figure 10 polymers-12-02072-f010:**
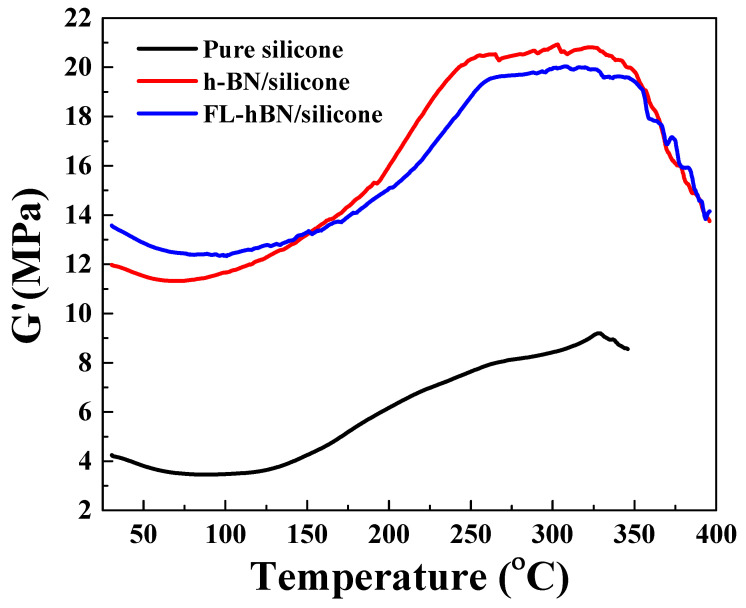
Dynamic mechanical analysis (DMA) curves of pure silicone, h-BN/silicone and FL-hBN/silicone composites.

**Table 1 polymers-12-02072-t001:** The physical properties of all the composites.

	Fraction (wt.%)	Density (cm^3^ g^−1^)	Specific Heat Capacity (J g^−1^ K^−1^)	Thermal Diffusivity (mm^2^ s^−1^)	Thermal Conductivity (W m^−1^ K^−1^)
Pure silicone	0	1.030	1.993	0.119	0.244
hBN/silicone	10	1.089	1.928	0.164	0.344
	20	1.141	1.386	0.226	0.357
	30	1.182	1.264	0.284	0.424
FL-hBN/silicone	10	1.060	1.944	0.203	0.418
	20	1.119	1.434	0.273	0.438
	30	1.170	1.381	0.319	0.515

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
