# Peer review of "Enhanced Thermal Conductivity of Silicone Composites Filled with Few-Layered Hexagonal Boron Nitride"

_polymers, 2020, doi:10.3390/polym12092072_

Round 1

Reviewer 1 Report

Overall, this manuscript demonstrated a method to achieve the few-layer h-BN with large aspect ratio, the verification methods and corresponding results almost make sense.  However, the authors need address some of issues list below.  In addition, I should mention there are quite a lot of careless in terms of writing throughout the main texts, which may annoy readers seriously. If this issue is not solved in the revised version, I wouldn’t accept it for publication.

Line 39: h-BN is an anisotropic material, having the out-of-plane thermal conductivity much lower than in-plane. Pleases refer to this paper: Communications Physics, volume 2, Article number: 43 (2019).

Lines 40-48:  All the references list seems related to the surface treatment, rather the aspect ratio. Please find the suitable reference and fit it.

Lines 49-51:  Though I agree with the conclusion, the argument is not persuasive, lacking some evident.

Line 52-53:  It is convective heat transfer, I don’t think this example is suitable here.

Line 59: Why graphene?  the paper is talking h-BN.

Line 74:  The draft never give introduction to jet cavitation method.  Is it novel method or existing method?

Line 78;  Full names of SEM AFM, etc,  are needed when they first appear.

Line 90:  It is wired that Figure S3 comes out early than Figure S1 and S2. And I notice that the SI material is barely used in the main text,  so why the SI material is useful?  Please just keep the used information in SI, drop those unused.

Line 90:  Why black,  shouldn’t be white?

Line 91: Should put some FL-h-BN image in the manuscript.

Line 118-120:  There is no explanation.  Why?

Line 125:  I guess it is “Figure 3a and 3d”. 

Line 137:  Does Fig 4(a) have the image of h-BN?

Line 161:  How is it observed? Which direction? Some literatures can be referred to : 1)ACS applied materials & interfaces 7 (23), 13000-13006 2) ACS Appl. Mater. Interfaces 2013, 5, 15, 7633–7640

Line 167:  The method of thermal conductivity hasn’t been introduced in Section 2

Line 172-173:  Nobody will understand this sentence unless realize there is a typo.

Line 173:  What percentage improved?

Line 176:  Same wt%,  but the density and specific heat are different between h-BN/Silicone,  FL-h-BN/Silicone.  Why?   And what the measurement method?

Author Response

Response to Reviewer 1:

Overall, this manuscript demonstrated a method to achieve the few-layer h-BN with large aspect ratio, the verification methods and corresponding results almost make sense. However, the authors need address some of issues list below. In addition, I should mention there are quite a lot of careless in terms of writing throughout the main texts, which may annoy readers seriously. If this issue is not solved in the revised version, I wouldn’t accept it for publication.

  1. Line 39: h-BN is an anisotropic material, having the out-of-plane thermal conductivity much lower than in-plane. Pleases refer to this paper: Communications Physics, volume 2, Article number: 43 (2019).

Response:

Thank you for reviewer’s comment. Yes. We have added this paper as an important reference in the revised manuscript.

[16] Yuan C.; Li J. H.; Lindsay L.; Cherns D.; Pomeroy J. W.; Liu S.; Edgar J. H.; Kuball M. Modulating the thermal conductivity in hexagonal boron nitride via controlled boron isotope concentration, Communications Physics, 2019, 2, 43, 1-8.

  1. Lines 49-51: Though I agree with the conclusion, the argument is not persuasive, lacking some evident.

Response:

Reviewer’s concern is right. We slightly change our conclusion in the revised manuscript. We hope reviewer could satisfy our revision.

According these references, we could understand that via introducing modified-BN as a filler into polymer matrix, the thermal conductivity could be enhanced. When the filler in polymer matrix has larger aspect ratio, the thermal conductivity of composites will be enhanced better based on the same weight percentage.

  1. Line 52-53: It is convective heat transfer, I don’t think this example is suitable here.

Response:

Thank you for reviewer’s comment. We have withdrawn the unsuitable reference in the revised manuscript.

  1. Line 59: Why graphene? the paper is talking h-BN.

Response:

We are sorry for the mistake. The necessary revision has been made in the revised manuscript.

  1. Line 74: The draft never give introduction to jet cavitation method. Is it novel method or existing method?

Response:

Thank you for reviewer’s kindly reminding. We have added the introduction and the corresponding references in the revised manuscript. Yes. It is an existing method. We have successfully synthesized few layer graphene, MoS2 and WS2 by this method.

The revised introduction session is shown as follow:

Recently, liquid exfoliation techniques of two-dimensional materials, such as high pressure homogenizer [34] and sonication [35], to delaminate these materials or using jet cavitation method to obtained few layer two-dimensional materials.[36] According to the above researches, these solvents are organic-based solvents, like NMP, DMF or acetone in the preparation process, even the surfactants including sodium carboxymethyl cellulose (CMC), non-ionic surfactant TWEEN®80 (TW80) or sodium dodecyl sulfate (SDS) are used. In our previous work, we have already synthesize few layer graphene, WS2 and MoSe2 by using jet cavitation method.[37-39]

  1. Skaltsas, T.; Ke, X.; Bittencourt, C.; Tagmatarchis, N. Ultrasonication induces oxygenated species and defects onto exfoliated graphene. The Journal of Physical Chemistry C, 2013, 117, 23272-23278.
  2. Liang, S.; Shen, Z.; Yi, M.; Liu, L.; Zhang, X.; Cai, C.; Ma, S. Effects of Processing Parameters on Massive Production of Graphene by Jet Cavitation. Journal of Nanoscience and Nanotechnology, 2015, 15, 2686-2694.
  3. Lin P.-C.; Wu J.-Y.; Liu, W.-R. Green and facile synthesis of few-layer graphene via liquid exfoliation process for Lithium-ion batteries. Scientific Reports, 2018, 8, 9766.
  4. Yeh Y.-Y.; Chiang, W.-H.; Liu, W.-R. Synthesis of few-layer WS2 by jet cavitation as anode material for lithium ion batteries. J. Alloys and Compounds, 2019, 775, 1251-1258.
  5. Wu Y.-C.; Liu W.-R. Few-layered MoSe2 ultrathin nanosheets as anode materials for lithium ion batteries. J. Alloys and Compounds, 2020, 831, 152074

  1. Line 78; Full names of SEM AFM, etc, are needed when they first appear.

Response:

Thank you for reviewer’s comments. SEM (Scanning Electron Microscopy) and AFM (Atomic Force Microscope) have added in the revised manuscript when they first appear.

The revised experimental session is shown as follow:

The thermal behaviors of the as-prepared materials were investigated using TGA analysis (Thermogravimetric analysis). The ramp rate was maintained 5°C/min under air atmosphere. The crystal phase structure and purity were determined by XRD (X-ray diffractometer) analysis using a D8 diffractometer (BrukerÒ) with monochromatic CuKα radiation. The operating voltage, current and wavelength (λ) were 40 kV, 30 mA, and 1.54060 Å, respectively. Diffraction data were recorded in the range (2θ) of 10–80°. The morphological natures of as-prepared samples were observed using SEM (Scanning Electron Microscopy, Hitachi S-4100) with electron mapping (EDS, Energy-dispersive X-ray spectroscopy). Atomic force microscope (AFM) images were captured by a Bruker Dimension Icon. The samples for AFM were prepared by dropping the dispersion directly onto freshly cleaved mica wafer with an injector. Brunauer–Emmett–Teller (BET) specific surface area was determined from N2 adsorption by using a Micromeritics TriStar 3000 (USA) analyzer at liquid nitrogen temperature.

  1. Line 90: It is wired that Figure S3 comes out early than Figure S1 and S2. And I notice that the SI material is barely used in the main text, so why the SI material is useful? Please just keep the used information in SI, drop those unused.

Response:

Thank you for reviewer’s suggestion. We have modified the SI material and drop some unused data in the revised manuscript.

  1. Line 90: Why black, shouldn’t be white?

Response:

The color of h-BN and FL-hBN is white. The black color is the background.

  1. Line 91: Should put some FL-h-BN image in the manuscript.

Response:

Thank you for reviewer’s good suggestion. We have put FL-hBN image in the revised manuscript. (Please see the revised Fig. 2)

Revised Fig. 2.

  1. Line 118-120: There is no explanation. Why?

Response:

    Thank you for reviewer’s comment. We have added the explanation in the revised manuscript.

The few-layer BN show a remarkably reduced diffraction intensity of XRD pattern indicates that the presence of thinner h-BN sheets and much less extended/ordered stacking in the c direction. The exfoliation may damage the crystal. Thus, the crystalline of h-BN may reduce after exfoliation.

  1. Line 125: I guess it is “Figure 3a and 3d”.

Response:

We are sorry for the mistake. We have corrected the mistake in the revised manuscript.

  1. Line 137: Does Fig 4(a) have the image of h-BN?

Response:

We don’t have AFM image of h-BN. Because the thickness of h-BN is too thick to test by AFM. We hope reviewer could satisfy with our response.

  1. Line 161: How is it observed? Which direction? Some literatures can be referred to : 1)ACS applied materials & interfaces 7 (23), 13000-13006 2) ACS Appl. Mater. Interfaces 2013, 5, 15, 7633–7640

Response:

Lattice orientation of the filler in the composites can be observed by checking the relative intensity of XRD pattern. The direction is (002) plane. Thank you for reviewer’s suggestion. These two references have been added in the revised manuscript.

  1. Yuan C.; Duan B.; Li L.; Xie B.; Huang M.; Luo X.B. Thermal Conductivity of Polymer-Based Composites with Magnetic Aligned Hexagonal Boron Nitride Platelets. ACS Applied Materials & Interfaces, 2015, 7, 23, 13000-13006.
  2. Lin Z.Y.; Liu Y.; Raghavan S.; Moon K.-S.; Suresh K; Sitaraman S.K.; Wong C.-P. Magnetic Alignment of Hexagonal Boron Nitride Platelets in Polymer Matrix: Toward High Performance Anisotropic Polymer Composites for Electronic Encapsulation, ACS Appl. Mater. Interfaces, 2013, 5, 15, 7633-7640.

  1. Line 167: The method of thermal conductivity hasn’t been introduced in Section 2

Response:

We are sorry for the mistake. The method of thermal conductivity has been added in Section 2 of the revised manuscript.

Thermal diffusivity (α) of the film was measured by laser flash thermal analyzer (LFA457 Micro Flash). The sample size was 1 cm x 1 cm x 0.2 cm.

  1. Line 172-173: Nobody will understand this sentence unless realize there is a typo.

Response:

We are sorry for the confusing. The necessary correction has been made in the revised manuscript.

  1. Line 173: What percentage improved?

Response:

The thermal conductivity was improved from 0.244 W/m K (bare silicone) to 0.515 W/ m K. The improved percentage is 2.3 times (230%).

The thermal conductivity of 30%FL-hBN/silicone (30 wt.% FL-hBN into silicone) composites was 0.515 W/m・K, which was higher than that of 30% hBN/silicone composites (0.424 W/m・K). Compared to pristine silicone, the thermal conductivity of FL-hBN/silicone composite enhanced 230% (2.3 times) by introducing 30 wt.% FL-h-BN.

  1. Line 176: Same wt%, but the density and specific heat are different between h-BN/Silicone, FL-h-BN/Silicone. Why?   And what the measurement method?

Response:

We understand the density and specific heat may different between h-BN/Silicone and FL-h-BN/Silicone. Because the nature of h-BN and FL-h-BN was quite different. It is difficult (or impossible) to control the same properties of these two sample. However, it is worth to notice that by using the exfoliation method, we could use less h-BN as a nano filler to enhance the thermal conductivity. The measurement method we used is by laser flash method.

Reviewer 2 Report

The current work provides a simple method to exfoliate the bulk h-BN into few-layer h-BN flakes and investigates its application in thermal conductivity. They used XRD, TEM and AFM techniques to characterize the obtained h-BN flakes. These characterization results indicated the synthesized h-BN flakes are few layers and have a lateral sized around ~2-5 μm. What is more, they also studied the influence of the amount of pristine and few-layer h-BN on silicone thermal conductivity. I think they obtained good results in the current experiment, but I hope they can resolve several questions before it is published.

  1. The exfoliated few-layer h-BN flakes were characterized via XRD. They said the peak intensity of few-layer h-BN flakes decreased, indicating the presence of thinner h-BN sheets, but the different amount of samples also will affect the intensity results. According to the results of other exfoliated h-BN sheets, if the bulk h-BN sample is exfoliated into few-layer sheets, the peak will appear reduced intensity and broadened width, this result is not shown in current results.
  2. The author should test the Raman spectrum of exfoliated h-BN flakes to ensure further that they obtained the few-layer h-BN flakes successfully.
  3. In the exfoliating process, whether groups, such as –OH will functionalize the h-BN sheets.
  4. For the thermal conductivity results, what mechanism induces the improvement of thermal?

Author Response

The current work provides a simple method to exfoliate the bulk h-BN into few-layer h-BN flakes and investigates its application in thermal conductivity. They used XRD, TEM and AFM techniques to characterize the obtained h-BN flakes. These characterization results indicated the synthesized h-BN flakes are few layers and have a lateral sized around ~2-5 μm. What is more, they also studied the influence of the amount of pristine and few-layer h-BN on silicone thermal conductivity. I think they obtained good results in the current experiment, but I hope they can resolve several questions before it is published.

  1. The exfoliated few-layer h-BN flakes were characterized via XRD. They said the peak intensity of few-layer h-BN flakes decreased, indicating the presence of thinner h-BN sheets, but the different amount of samples also will affect the intensity results. According to the results of other exfoliated h-BN sheets, if the bulk h-BN sample is exfoliated into few-layer sheets, the peak will appear reduced intensity and broadened width, this result is not shown in current results.

Response:

Reviewer’s comment is right. We should use FWHM instead of intensity. We have calculated Lc and La in Table S1. As you can see, the diffraction peak (002) of h-BN become broaden after exfoliation. The correspond data and discussion have been added in the revised manuscript and supporting information.

Table S1. FWHM, Lc and La of h-BN and FL-hBN.

FWHM(002)(⁰)

Lc (002)(Å)

La (100)(Å)

h-BN

0.3191

256.2

228.8

FL-hBN

0.4130

197.9

213.5

  1. The author should test the Raman spectrum of exfoliated h-BN flakes to ensure further that they obtained the few-layer h-BN flakes successfully.

Response:

Thank you for reviewer’s suggestion. Raman, indeed, is a very powerful tool to identity thickness and quality of 2 dimensional materials. However, in this study, our h-BN is not single layer. Thus, it is difficult to identify our material by Raman. That is the reason why we use SEM, TEM, AFM and BET. We hope reviewer could understand it.

  1. In the exfoliating process, whether groups, such as –OH will functionalize the h-BN sheets.

Response:

In fact, our exfoliation process is a mild and physical exfoliation process. We believe functional group, such –OH, may not functionalize the h-BN sheets.

  1. For the thermal conductivity results, what mechanism induces the improvement of thermal?

Response:

Please kindly see the following Figure. It may be useful to explain the mechanism why we could improve thermal conductivity by using few layer hBN in silicone.

Thanks for your assistance.

Sincerely yours,

Wei-Ren Liu,

Professor,

Department of Chemical Engineering,

Chung Yuan Christian University

Round 2

Reviewer 1 Report

No further comments